# Chitosan-Based Polymeric Nanoparticles as an Efficient Gene Delivery System to Cross Blood Brain Barrier: In Vitro and In Vivo Evaluations

**DOI:** 10.3390/ph17020169

**Published:** 2024-01-29

**Authors:** Ishaq N. Khan, Shiza Navaid, Walifa Waqar, Deema Hussein, Najeeb Ullah, Muhammad Umar Aslam Khan, Zakir Hussain, Aneela Javed

**Affiliations:** 1MIT Media Lab, Massachusetts Institute of Technology, Cambridge, MA 02139, USA; ishaqene101@gmail.com; 2Cancer Cell Culture & Precision Oncomedicine Lab, Institute of Basic Medical Sciences, Khyber Medical University, Peshawar 25100, Pakistan; najeebwazir@hotmail.com; 3School of Chemical and Materials Engineering, National University of Sciences and Technology, Islamabad 44000, Pakistan; shizan995@gmail.com; 4Atta-ur-Rahman School of Applied Biosciences, National University of Sciences and Technology, Islamabad 44000, Pakistan; walifawaqar@yahoo.com; 5Neurooncology Translational Group, King Fahd Medical Research Center, King Abdulaziz University, P.O. Box 80216, Jeddah 21589, Saudi Arabia; deemah@hotmail.com; 6Department of Mechanical and Industrial Engineering, Qatar University, Doha 2713, Qatar; umar007khan@gmail.com; 7Biomedical Research Center, Qatar University, Doha 2713, Qatar

**Keywords:** blood brain barrier, brain cancer, gene therapy, natural polymeric nanoparticles, transfection, brain tumor targeting

## Abstract

Significant progress has been made in the field of gene therapy, but effective treatments for brain tumors remain challenging due to their complex nature. Current treatment options have limitations, especially due to their inability to cross the blood-brain barrier (BBB) and precisely target cancer cells. Therefore options that are safer, more effective, and capable of specifically targeting cancer cells are urgently required as alternatives. This current study aimed to develop highly biocompatible natural biopolymeric chitosan nanoparticles (CNPs) as potential gene delivery vehicles that can cross the BBB and serve as gene or drug delivery vehicles for brain disease therapeutics. The efficiency of the CNPs was evaluated via in vitro transfection of Green Fluorescent Protein (GFP)-tagged plasmid in HEK293-293 and brain cancer MG-U87 cell lines, as well as within in vivo mouse models. The CNPs were prepared via a complex coacervation method, resulting in nanoparticles of approximately 260 nm in size. In vitro cytotoxicity analysis revealed that the CNPs had better cell viability (85%) in U87 cells compared to the chemical transfection reagent (CTR) (72%). Moreover, the transfection efficiency of the CNPs was also higher, as indicated by fluorescent emission microscopy (20.56% vs. 17.79%) and fluorescent-activated cell sorting (53% vs. 27%). In vivo assays using Balb/c mice revealed that the CNPs could efficiently cross the BBB, suggesting their potential as efficient gene delivery vehicles for targeted therapies against brain cancers as well as other brain diseases for which the efficient targeting of a therapeutic load to the brain cells has proven to be a real challenge.

## 1. Introduction

The advent of gene therapy in the early 1990s raised expectations for brain tumor therapies and clinical trials in patients; however, the therapeutic benefits were not convincing, primarily due to the inefficient delivery of therapeutic payload drugs and genes to cancer cells [1,2]. Thus, finding a vehicle or carrier that efficiently delivers a specific gene or gene segment to tumor tissue remains crucial. Designing a desirable vector is highly important for effective gene therapy, which usually needs robust and constant gene expression in targeted cells, which circumvents transgene-related side effects such as overexpression, immunogenicity, or off-targets [3]. Delivery vectors such as viral vectors, non-polymeric nanoparticles (NPs), and polymeric NPs have been used to deliver the therapeutic payload in GBM (glioblastoma multiforme) and LGGs (Lower-grade Gliomas) [4]. Viral vectors have been used as gene-delivery vehicles but have not yet been clinically approved due to their manufacturing challenges, high cost, immunogenic and inflammatory responses, oncogenic mutations, and limited loading capacity [5]. Non-viral delivery strategies, including non-viral vectors (polymeric and non-polymeric), offer alternative approaches for overcoming the barriers of gene delivery. Carbon nanoparticles engineered to deliver chemotherapy drugs across the blood-brain barrier and mark tumor cells with fluorescence in mice have proven successful in using nanomaterial for brain cancer therapeutics [6].

Glioblastoma multiform (GBM), an aggressive brain cancer, can invade surrounding brain tissues, resist therapy, and reoccur [7]. Despite tremendous efforts to develop diagnostic tools and therapeutic avenues, the treatment of brain tumors remains a formidable challenge in the field of neuro-oncology. Although complete resection is the most effective current treatment option for inoperable tumors, radiotherapy and chemotherapy are often recommended [8]. Unfortunately, many side effects and deficits are often associated with the latter approach, and therapeutic delivery is an insufficient solution due to its inability to cross the blood-brain barrier (BBB), which is required to cause adequate destruction of malignant cells [9].

Scientists are trying to utilize nanoparticles to regulate cellular microenvironments, enhance the cells’ efficiency, and deliver drugs to the brain [10]. Considerations regarding the various aspects of different nanoparticles are vital for applying them effectively in clinical therapies [11]. Moreover, to increase the efficiency of the transfecting nanoparticles, using naturally occurring materials is considered a better option. Polysaccharides are biopolymers that have gained much attention for being biocompatible, safe, biodegradable, and efficient [12]. Chitosan (CS), a polysaccharide obtained from crustaceans’ exoskeletons, has a structural resemblance with hyaluronic acid (HA), which is a central component of the brain [13]. Chitosan is composed of repeating units of D-glucosamine [14]. CS-based formulations are commonly used for gene delivery because of their exceptional properties, including biocompatibility, natural antibacterial, neuroprotective, and anti-inflammatory properties [15]. Furthermore, the in vivo enzymatic degradation of chitosan into N-glucosamine is performed through enzymes like chitosanase and lysozyme, which are endogenously present in the human body [16].

Indeed, there is a need to devise new biocompatible delivery tools that can easily transfect tumor cells with decreased cytotoxicity and immunogenicity and can replace conventional gene delivery vehicles. This current study thus focused on the evaluation and use of chitosan nanoparticles (CNPs) coated with the gene of interest (GFP in this case) to transfect the U-87 MG (Human Glioblastoma) cell line. The uptake of the GFP-conjugated CNPs was evaluated through in vitro and in vivo assays using fluorescent microscopy and fluorescence-activated cell sorting (FACS) techniques. GFP was used as a control as it can be easily visualized and tracked for in vitro transfections crossing the cell membrane and in vivo crossing of the BBB through various techniques utilized in many studies, including FACS and fluorescent microscopy. The successful transfer of the desired protein into the cell line (in vitro) as well as the (in vivo) brains of the animal models indicates that the chitosan nanoparticles can be used as potential target-specific delivery vehicles or nano carriers against several brain-instigated pathologies, including brain tumors for which crossing the blood brain barrier for therapeutics is a significant treatment challenge.

## 2. Results

### 2.1. Characterization of Chitosan-GFP Nanoparticles

#### 2.1.1. Fourier-Transform Infrared Spectroscopy FTIR

Following the chitosan-based nanoparticle preparation (Figure 1), the FTIR spectrum of the chitosan-GFP nanoparticles was performed to confirm the formation of chitosan-GFP complexes and was obtained in the range of 4000 to 400 cm^−1^. As per spectrum analysis, the peak at 1650 cm^−1^ indicates the presence of NH_2_ vibration, and 1561 cm^−1^ is of a carbonyl-ring-stretching vibration from chitosan [17]. The absorbance bands at 1642.50 cm^−1^ and 1633.93 cm^−1^ are from nucleotides stretching and in-plane vibrations from the DNA molecule, while the characteristic peaks at 1236 cm^−1^ of phosphate ester and at 963 cm^−1^ result from a 2′ endo deoxyribose conformation [18]. The chitosan-GFP nanoparticles showed characteristic peaks at 891 cm^−1^ of the pyranose ring and 1561 cm^−1^ of the carbonyl ring vibrations, indicating the complex formation Figure 2I.

#### 2.1.2. Scanning Electron Microscopy (SEM)

The morphology of the chitosan nanoparticles and chitosan-GFP nanoparticles was investigated through scanning electron microscopy, as shown in Figure 2II. Both the chitosan nanoparticles and the chitosan-GFP nanoparticles can be observed from the images as spherical. The size of the chitosan nanoparticles was within the range of 89–132 nm. However, the size of the chitosan-GFP nanoparticles was ~260 nm.

#### 2.1.3. Dynamic Light Scattering (DLS)

The spectra of the size distribution and zeta potential for the chitosan and chitosan-GFP nanoparticles are shown in Figure 2III. The average size distribution of the chitosan nanoparticles was 96.34 nm (Figure 2III(A)) with 17.9% intensity, and a second peak at 316.3 nm was observed with an intensity of 82.1%. The zeta potential of the chitosan nanoparticles was 28.8 mV (Figure 2III(C)). In comparison, the average size of the chitosan-GFP nanoparticles was 289 ± 26.82 nm (Figure 2III(B)) with a zeta potential of 10.6 mV (Figure 2III(D)).

### 2.2. Amplification of FAM Plasmid DNA Using Colony PCR

The isolation of FAM26F DNA was evaluated using the colony PCR. Twelve random colonies were picked from cultured plates for performing colony PCR, followed by agarose gel electrophoresis. The results confirmed the isolation of the desired plasmid DNA of interest via a band size of 220 bp, as shown in Figure 1I).

### 2.3. In-Vitro Cytotoxicity Analysis

The cytotoxicity of the chitosan nanoparticles and the chitosan-GFP nanoparticles was evaluated via an in vitro cytotoxicity assay, as shown in Figure 3II, to evaluate their suitability for the transfection of cells. It was found that neither the chitosan nanoparticles alone, nor the chitosan-GFP nanoparticles, were cytotoxic towards the U87 cells. Cell viability was found to be 85% with CS-NP and 80% with DNA-conjugated CS-NP, which depict the efficacy of this complex for utilization as a transfection reagent for gene delivery. CTR, the commercially available transfection reagent, was also used for comparative analysis, showing 72% viability alone and 66% cell viability with the conjugated DNA. Our results thus indicate that the chitosan nanoparticles transfect brain cells with reduced cytotoxic effects (*p* > 0.05) as compared to the commercially available reagent, CTR, making them ideal for use in transfection for therapeutic purposes.

### 2.4. Fluorescence Emission Microscopy

a.In vivo payload delivery efficiency analysis:

The biodistribution of intra-peritoneally injected CS nanoparticles labeled with GFP was tracked in the brain to assess the CNPs’ capability of crossing the BBB. The fluorescence intensity of the green signals indicated the number of positive cells for GFP within the brain. The number of GFP-positive cells was found to be 14.2%, 11.2%, and 8.7% in the three-extracted brain from the CNPs-treated Balb/c mice (*p* > 0.05, * ANOVA: *p* = 0.197; *n* = 5; F crit (3.8) > F (1.8)), Figure 4B). It was promising that the CNPs were identified as crossing the BBB and reaching the brain.

b.In vitro transfection efficiency analysis

The U87 and HEK293 cell lines transfected with the chitosan-GFP nanoparticles were observed under a fluorescent microscope (Nikon Eclipse 80i, Nikon, Tokyo, Japan) using two filter sets: blue for nuclear staining and green for the expression of the GFP (Green Fluorescent Protein) tag. The commercially available transfection reagent (CTR) was used as a positive control. A quantitative increase in the green fluorescence signal was observed for both transfection agents after 48 h as compared to the untreated control. Green fluorescence was observed in both cells transfected with the chitosan-GFP nanoparticles and the CTR-transfected cells, ensuring successful transfection. The transfection efficiency of the CTR on HEK293 and U87 cell lines was 18.92% and 17.79%, respectively. In contrast, the transfection efficiency of the chitosan-GFP nanoparticles was 27.41% and 20.56% in the HEK293 and U-87 cell lines, respectively, Figure 5I. Moreover, the number of viable cells after transfection with the CTR was less than the number of viable cells after transfection with the CS-NPs, thus indicating the CS-NPs to be more efficient and less cytotoxic to the cells, as shown in Figure 6.

### 2.5. Fluorescent Activated Cell Sorting (FACS) Analysis for Transfection Efficiency

The transfection efficiency of the chitosan nanoparticles was also analyzed using FACS by evaluating the expression of the green fluorescent protein (GFP) in the transfected cells. In the HEK293 cell line, the chitosan nanoparticles showed 21% transfection efficiency, while the CTR showed 56% transfection efficiency. Parallel to the fluoresce microscopy, the CNPs showed higher transfection efficiency (53%) than the commercially available transfection reagent CTR (27%) on U-87 cell lines, as shown in Figure 5.

## 3. Discussion

Neurological disorders affect more than one billion people worldwide, and no effective therapies are currently available [18]. Several promising experimental drug candidates fail successive clinical trials due to insufficient delivery across the BBB and the subsequent expelling of the therapeutic payload through efflux pumps (ABC transporters), which leads to severe side effects such as diabetes, seizures, and toxicity. Innovative technologies, including nanocarriers (NC) with “enhanced permeability and retention” (EPR), are being explored. These therapies ensure targeted and efficient drug delivery across the complex BBB for optimal therapeutic loading of genes/drugs into the brain to treat central nervous system (CNS) disorders [19].

Chitosan, a commonly used biopolymer for gene and drug delivery, has been used to target various cells because of its good biocompatibility. It is considered an efficient and safe transfection reagent and can be degraded by enzymes in the human body. The presence of amino groups in the chitosan molecule gives the polymer a positive charge that aids in formin complexes with the negatively charged DNA molecules. Chitosan nanoparticles have shown good transfection efficiencies via in vitro studies performed on various cell lines, e.g., HEK293, MG63 cells, MSCs, HeLa cells [20], and A549 cells [21]. However, few transfection studies of chitosan nanoparticles on brain cancer cells have been conducted [22]. The scarcity of transfection agents that effectively cross the blood-brain barrier necessitates exploring efficient gene delivery vehicles.

Nanomedicines are considered non-invasive and are one of the most promising approaches for treating neurological disorders [11,21], and using nanoparticles has been explored for treating various diseases [3]. Glioblastoma multiform (GBM) is a brain-cancer-forming diffused tumor that resists therapy [10] and hence requires rigorous efforts in the wake of innovative therapeutic strategies. Systemically administered nanomedicine strategies can be divided functionally into (a) stimuli-responsive strategies, (b) tissue-microenvironment-reprogramming strategies, (c) transposable nanomedicine, and (d) immuno-oncological nanomedicine [23]. Chitosan polymer offers advantages, including muco-adhesion and functionalization ease. Chitosan-based nanocarriers (CsNCs) act by establishing ionic interactions with the endothelial cells, thus accelerating the BBB crossing through adsorptive transcytosis. CsNCs can further be modified due to their reactive amino and hydroxyl groups. In the attachment/binding of ligands such as antibodies or lipids, CsNCs express a boosted passage across the BBB [24]. Th neural extra-cellular matrix (ECM) in human brains is composed of proteoglycans and hyaluronic acid (HA) [25]. As chitosan is structurally similar to HA, chitosan nanoparticles are expected to be an excellent biological source for entering the brain cells. This study used chitosan nanoparticles to transfect the U-87 GBM cell line in order to evaluate their potential as gene delivery vehicles for treating brain cancers. By analyzing the CNPs via in vitro and in vivo studies, it was found that the CNPs are less toxic, possess more potential to transfect the brain cancer cell lines, and hold the ability to cross the BBB.

According to previous studies, chitosan gives characteristic peaks at 1657 cm^−1^ of NH_2_ vibration, 1564 cm^−1^ of carbonyl-stretching vibration, and 1070–1029 cm^−1^ of stretching vibrations of C–O of the pyranose ring [17]. DNA gives absorbance bands at 1602 cm^−1^ for adenine, 1650 cm^−1^ for thymine, 1684 cm^−1^ for guanine, and 1481 cm^−1^ for cytosine. The characteristic peaks at 1236 cm^−1^ of phosphate ester and 963 cm^−1^ result from 2′ endo deoxyribose conformation [26]. Our results show similar results in the FTIR spectrum of the chitosan-GFP nanoparticles synthesized in this study with characteristic peaks at 891 cm^−1^ of the pyranose ring and 1561 cm^−1^ of the carbonyl ring, indicating a successful complex formation between the chitosan nanoparticles and DNA. A few characteristic peaks of chitosan and DNA overlap in the nanoparticle formation. Furthermore, DNA’s vibration bands were almost unchanged as compared to the simple DNA. Hence, the FTIR spectrum suggests that DNA protects ꞵ-conformation in the chitosan-GFP complex. Other studies suggesting a good complex formation between chitosan and DNA molecules have reported similar spectrums and characteristic peaks [27,28], thus bolstering our results.

The SEM images indicated that the average chitosan-GFP nanoparticle size was about 260 nm and had a nearly spherical shape. The average size distribution of the chitosan and chitosan-GFP nanoparticles was ~316 nm and 289 nm, respectively. The decrease in size after complex formation can be attributed to the size range for chitosan-GFP nanoparticles, which is similar to the studies conducted by Zhao et al. and Fernandez et al. [29,30]. The zeta potential is 28.8 mV for the chitosan nanoparticles and 10.6 mV for the chitosan-GFP nanoparticles. The size and zeta potential of the chitosan nanoparticles decreased after the conjugation with the plasmid DNA. The formation of the chitosan-GFP nanoparticles led to size shrinkage as well as a drop in the zeta potential, and it is due to the anionic nature of DNA that the zeta potential dropped from 28.8 mV to 10.6 mV after complex formation with the cationic chitosan molecule [31].

The cytotoxic effects of the chitosan nanoparticles and the chitosan-GFP nanoparticles were evaluated and compared to the commercially available CTR-6 transfection reagent on the U87 cell line via MTT assay. Our results demonstrated that the chitosan-GFP nanoparticles were relatively less cytotoxic than the CTR-6 and caused less than 20% cell death, as opposed to the significant decrease of 37% in cell viability using CTR-6. Other studies have also reported the cell viability after transfection with chitosan complexes to be between 60–90%, which is consistent with our results showing 80% cell viability [31,32]. Chitosan has frequently been reported as a biocompatible agent inducing lower cytotoxicity in cell lines [33], which is our result.

The DNA plasmid used to make complexes with the chitosan nanoparticles was tagged with GFP (Green Fluorescent Protein), which can help monitor gene expression quantitatively. Thus, the transfection studies on the HEK293 293 cell line and the U-87 cell line using the CTR-6 (positive control) and the chitosan-DNA nanoparticles were analyzed using fluorescent microscopy and FACS analysis. The presence of the GFP gene in the vector acted as the reporter, and the expression of the GFP ensured successful delivery.

The fluorescent microscopy results revealed that after 48 h, CTR-6 could transfect 18.92% and 17.79% of cells in the HEK293 cell line and the U-87 cell line, respectively. The chitosan-GFP nanoparticles transfected 27.41% and 20.56% of cells in the HEK293 cell line and the U-87 cell line, respectively. The transfection efficiency of the chitosan-GFP nanoparticles in both cell lines was greater than that of the commercially available transfection agent. Other studies have also shown that chitosan nanoparticles efficiently transfect various cells, including the HEK293 cell line and U87 cells [20,21].

FACS analysis quantitively showed that the transfection efficiency of the chitosan-GFP nanoparticles in U-87 cells was 53% after 24 h of transfection as compared to the 27% efficiency of CTR-6, a commercially available transfection reagent. It indicates the effectiveness of using chitosan-GFP nanoparticles to transfer genetic material in the brain cells (U-87 cell line). The use of chitosan as a suitable transfection agent has also been reported by Fernandez et al. as having 27% efficiency for airway epithelial cells [31], a result that strengthens our study results which found enhanced transfection efficiency. Our study demonstrates the reliability and efficiency of chitosan nanoparticles for use as gene delivery vehicles, specifically for brain cells. It opens new ventures of research for devising novel therapeutic strategies for brain disorders that use chitosan nanoparticles for gene therapies.

## 4. Materials and Methods

### 4.1. Synthesis of Chitosan Nanoparticles

The chitosan nanoparticles were prepared using the complex coacervation method described by MacLaughlin et al. [34]. The chitosan (low molecular weight from Sigma Aldrich) was dissolved in a 1% acetic acid (Sigma Aldrich, Merck, Darmstadt, Germany) solution to prepare a 2% chitosan solution and was diluted to make a 0.2% chitosan stock solution, followed by vacuum filtration. The overall preparation plan is depicted in Figure 1.

### 4.2. Amplification of Plasmid DNA (Enclosing GFP)

a.Competent DH5α (*Escherichia coli*) Cell Preparation and Transformation

The preparation of the competent cells was performed as described by Cohen et al. [35,36]. The DH5α strains of *E. coli* (Thermo Fisher Scientific, Inc., Waltham, MA, USA) were used for the vector transformations. For the transformation of competent cells, 1 µL of the GFP (Green Fluorescent Protein)-tagged plasmid DNA (CAT#: RG222648) was added to 100 µL of competent DH5α cells, mixed gently, and left on ice for 40 min. This was followed by heat shock at 42 °C (Memmert, GmbH, Buchenbach, Germany) for 90 s, after which it was placed on ice. The cells were added to LB Agar medium plates containing antibiotics (Thermo Fisher Scientific, Inc., Waltham, MA, USA) and incubated at 37 °C for 16 h, after which the transformed colonies appeared.

b.Miniprep for Alkaline Lysis

A single colony of transformed DH5α cells was mixed in LB media containing antibiotics and incubated for 16 h (overnight) at 37 °C (Memmert, GmbH, Buchenbach, Germany) with constant shaking. The supernatant was discarded to obtain the bacterial cell pellet. It was then resuspended in cold solution Ⅰ (Resuspension Buffer) and placed on ice. Solution Ⅱ (Lysis Buffer) was added to it and mixed, and the mixture was placed on ice. Then, 150 µL of cold solution Ⅲ (Neutralization Buffer) was added and mixed gently with bacterial lysate. The mixture was centrifuged and the transferred supernatant was placed in a fresh Eppendorf tube, to which an equal volume of chloroform (Sigma Aldrich, Merck, Darmstadt, Germany) and phenol (Sigma Aldrich, Merck, Darmstadt, Germany) (1:1) was added. The supernatant was then vortexed and centrifuged. The DNA was precipitated out by adding two volumes of ethanol (Sigma Aldrich, Merck, Darmstadt, Germany). After air-drying the pellet, 50 µL of Tris EDTA Buffer (Sigma Aldrich) (pH 8.0) containing DNase-free RNase (Thermo Fisher Scientific, Inc., Waltham, MA, USA) was added. The DNA extracted from the transformed bacterial colonies was quantified using Thermo-scientific Nanodrop Products 2000.

c.Colony PCR of the transformed colonies

To validate the isolation of the plasmid DNA (GFP tagged), colony PCR was performed. The transformed colonies were picked and suspended in nuclease-free water (Thermo Fisher Scientific, Inc., Waltham, MA, USA) [Lyse colonies at 95 °C for 10 min; plasmid DNA to be released in the nuclease-free water]. PCR was performed under the following conditions in a PCR machine (Miniamp, Thermofisher Scientific): the denaturation temperature was kept at 95 °C for 5 min, and then 30 cycles were performed with the annealing temperature at 95 °C for 45 s, 62 °C for 45 s, 72 °C for 45 s, and the final extension at 72 °C for 7 min. To confirm amplification, agarose gel electrophoresis was carried out.

### 4.3. Preparation of Chitosan-Tagged GFP Nanoparticles

A 10 mM sodium acetate (Sigma Aldrich, Merck, Darmstadt, Germany) buffer was prepared with pH 5.5 and used to make a 0.7% (*w*/*v*) chitosan solution. GFP plasmid DNA (0.1 µg/µL) was added to a 20 mM sodium sulfate buffer (pH 5.5). Equal volumes of both buffers containing chitosan and plasmid DNA were heated separately at 55 °C, mixed, and vortexed. The nanoparticles were then allowed to self-assemble with plasmid DNA at room temperature for 1 h.

### 4.4. Characterization of the Chitosan-GFP Nanoparticles

a.Fourier-Transform Infrared Spectroscopy (FTIR)

Fourier-Transform Infrared Spectrum (FTIR) was recorded over the range of 4000 to 400 cm^−1^ on the liquid sample containing the chitosan-GFP nanoparticles to confirm the functional groups and the interactions present in the chitosan-GFP nanoparticles, as well as to obtain the infrared absorption spectrum of the nanoparticles. The spectrum of the CS-GFP nanoparticles was recorded using the KBr pellet technique [37].

b.Scanning Electron Microscopy (SEM)

The structural analysis of the chitosan-GFP nanoparticles was performed using a Jeol JSM-6490A Analytical Scanning Electron Microscope (SEM). A drop of colloidal suspension containing the chitosan-GFP nanoparticles was placed on a square-cut glass slide. The sample was gold-sputtered and examined at an acceleration voltage of 10 kV at different resolutions and magnifications [16].

c.Dynamic Light Scattering (DLS)

Dynamic light scattering and non-invasive backscattering (DLS-NIBS) at 25 °C with automatic attenuator settings were used to measure the chitosan-GFP nanoparticle size distribution. The zeta potential was measured using a zeta dip cell in automatic mode. The parameters were determined using a Malvern zeta sizer (Malvern Instruments, Worcestershire, UK) [31].

### 4.5. In-Vitro Studies on Cell Lines

a.Cell Culture and In-vitro Cytotoxicity Assay

The MG-U87 cell lines (a kind gift from Norah Defamie, Norah Defamie Pôle Biologie Santé, Université de Poitiers, Poitiers, France) and the HEK293293 cells were allowed to grow in 10% FBS (fetal bovine serum) (Gibco, Thermo Fisher Scientific, Waltham, MA, USA) supplemented with DMEM (Dulbecco’s Modified Eagle Medium) (Gibco, Thermo Fisher Scientific, USA) and 500µL penicillin-streptomycin (Gibco, Thermo Fisher Scientific, Waltham, MA, USA) in a T-25 flask (NestBio, Cambridge, MA, USA) a with cell density of 4.0 × 10^6^ cells, in a CO_2_ incubator (Memmert, GmbH, Buchenbach, Germany) at 37 °C. The cells were allowed to grow until they reached 70% confluency. The cytotoxicity analysis of the synthesized chitosan-GFP nanoparticles was evaluated using in vitro cytotoxicity assay, as described [38]. A total of 1 × 104 viable U87 cells/well were seeded into a 96-well plate in 100µL of cell culture media and incubated overnight for 24 h at 37 °C in a 5% CO_2_ incubator. The CS-GFP nanoparticle complexes were incubated for 30 min at 37 °C. The plasmid DNA (GFP tagged), chitosan nanoparticles, chitosan-GFP nanoparticles, and CTR and CTR-DNA complexes were mixed in DMEM, added to the cells, and incubated at 37 °C. After 24 h incubation, the medium was replaced with MTT reagent (Phytotech Laboratories, Lenexa, KS, USA) (dissolved in PBS) and incubated for 3–4 h with the same parameters. After purple precipitates of formazan were observed, the solution was carefully aspirated, and 100 µL of DMSO was added to dissolve the formazan crystals formed by the live cells and kept in the dark at room temperature for an hour.

The cell viability was assessed through the absorbance of the formazan crystals, which was measured at 570 nm using a microplate reader (SAFIRE II, Tecan Group Ltd., Männedorf, Zurich, Switzerland). The percentage of cell viability was calculated using the following equation:
Cell viability (%) = OD570(sample)/OD570(control) × 100

where the OD570 (sample) is the measurement of absorbance from the wells treated with the plasmid DNA, chitosan nanoparticles, CTR transfection reagent, and the nanoparticles and CTR conjugated with DNA, while the OD570 (control) was the measurement of absorbance from the wells treated with the DMEM/cells only.

b.Transfection and Fluorescent Microscopy

The transfection efficiency of the chitosan-GFP nanoparticles was evaluated on the U87 cell line and the HEK293293 cells. The cells were cultured 3 days prior to transfection. Cells were washed with 1X PBS, trypsinized, and seeded in a six-well plate with a seeding density of 2 × 10^6^ per well. After 60–70% confluency, the media was removed, and the cells were washed with a 1X PBS solution. One percent serum-supplemented media (9 mL DMEM + 1 mL FBS) and chitosan-GFP nanoparticles containing 1.8 µL of DNA (GFP tagged), were added to the six-well plate and incubated at 37 °C in a CO_2_ incubator for 48 h. CTR 6 was used for the positive control, while for the negative controls, 1% supplemented media was added to the cells without adding the transfection reagent. After 48 h of incubation, cells were washed with 1X PBS and fixed with a 4% formaldehyde solution for 10 min. The coverslips with fixed cells were washed with PBS and stained with 1 µg/mL of DAPI solution (Thermo Fisher Scientific, Waltham, MA, USA) (nuclear stain) for 10 min. Following the DAPI staining, the coverslips were fixed on slides with the addition of mounting media and sealed with nail varnish. The slides of the transfected cells were then viewed under a fluorescence microscope (Nikon Eclipse 80i) to confirm the transfection and expression of GFP in the treated cells.

### 4.6. In-Vivo Transfection Studies on BALB/c Mice Model

To validate the entry of CS-GFP nanoparticles through the BBB in vivo, transfection studies were carried out on mice. A group of three 7-week-old BALB/c mice were given 200 µL of the CS-GFP nanoparticle solution via intraperitoneal injection. After 24 h, the mice were euthanized via chloroform, and the brains were carefully harvested by dissection. The brain samples were fixed in a 4% paraformaldehyde (PFA) solution (Sigma Aldrich, Merck, Darmstadt, Germany) for 24 h at 4 °C, followed by dehydration protocol. The dehydration was performed using various increasing concentrations of ethanol (30%, 50%, 70%, 80%, and 100%) for 30 min, followed by a 10 min treatment with xylene. The dehydrated samples were fixed in paraffin wax (Thermo Fisher Scientific, Waltham, MA, USA), and slides were prepared using a microtome in which each tissue slice thickness was 5 µm. The tissues on slides were deparaffinized and rehydrated, followed by a 10 min treatment with a 1 µg/mL DAPI solution for nuclear staining.

### 4.7. Fluorescent Activated Cell Sorting (FACS) Analysis

Flow cytometry also assessed the transfection efficiency of the transfection reagents, i.e., the CTR 6 and chitosan nanoparticles. The transfected cells to be analyzed were grown until 70% confluent, washed with 1X PBS, trypsinized, collected in an FBS supplemented medium, and brought to single-cell suspension. To determine the cell viability and GFP expression analysis, the cells were sorted on a BD FACScan flow cytometer. Morphological gating was performed, and the debris was removed based on forward and side scatter. The cells (events) were studied for their GFP expression. Flow cytometry data analysis was carried out using CellQuest BD FACScan.

## 5. Conclusions

The use of chitosan nanoparticles for gene delivery in brain cells has not been widely explored. Our study results demonstrate that the chitosan-GFP nanoparticles can be efficient gene delivery vehicles in the brain cells. U87 MG GBM cells, after 24–48 h of transfection with chitosan nanoparticles, continued to proliferate and efficiently express GFP, revealing that chitosan has no or minimal cytotoxic effect. In vivo studies indicated that the CNPs have the potential to cross the BBB. Chitosan nanoparticles are easy to manufacture and cost-efficient; hence, they can be considered promising gene delivery vehicles for transfecting brain cells and for using in gene therapy treatments of various CNS disorders and brain cancers. They can also be functionalized to make the nanoparticles specific for targeted cells.

The use of chitosan nanoparticles for gene delivery in brain cells has not been widely explored. Our study has demonstrated that chitosan-GFP nanoparticles can be efficient gene delivery vehicles in brain cells. The glioblastoma MG-U87 cells continued to proliferate and efficiently express GFP after 24–48 h of transfection with chitosan nanoparticles, which indicates that chitosan has no or minimal cytotoxic effect. In vivo studies also indicated that the CNPs have the potential to cross the BBB. Chitosan nanoparticles are easy to manufacture and cost-efficient; hence, they can be considered suitable gene delivery vehicles for transfecting the brain cells and using in gene therapy treatments of various CNS disorders and brain cancers. They can also be functionalized to make the nanoparticles specific for targeted cells. Chitosan nanoparticles might emerge as one of the most attractive formulations for brain delivery due to their efficient delivery, biocompatibility, and biodegradability.

## Figures and Tables

**Figure 1 pharmaceuticals-17-00169-f001:**
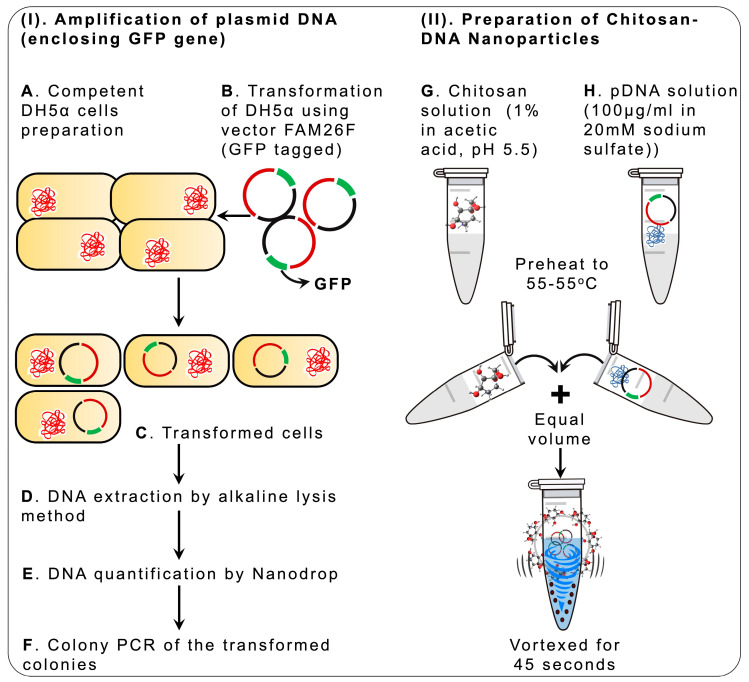
Schematic diagram of the chitosan DNA nanoparticles preparation. (**I**) Amplification of plasmid DNA (enclosing GFP gene). (**II**) Preparation of the chitosan-DNA nanoparticles.

**Figure 2 pharmaceuticals-17-00169-f002:**
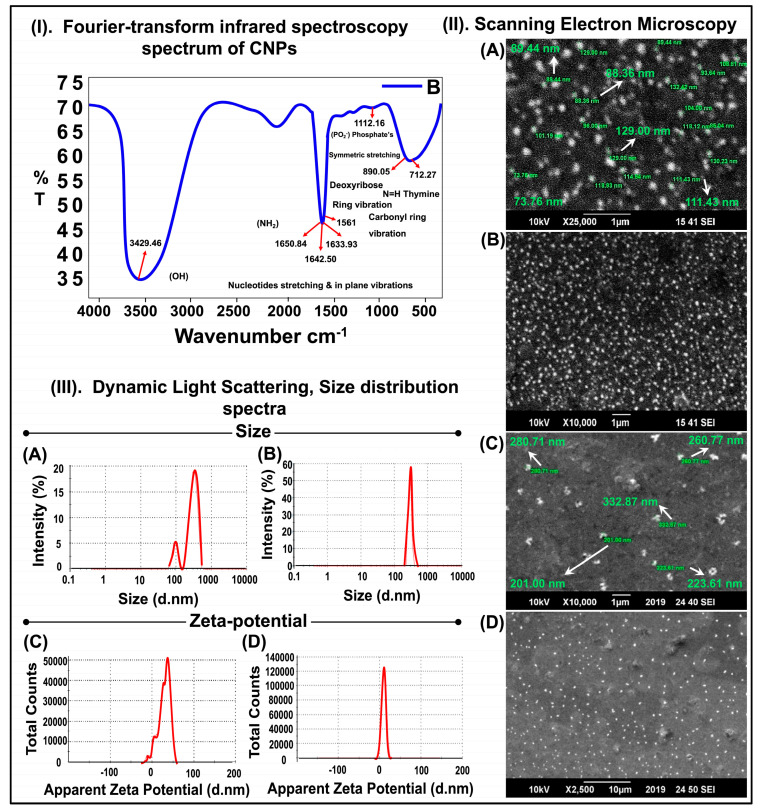
Characterization of chitosan-GFP nanoparticles: (**I**) FTIR spectrum of chitosan-GFP nanoparticles at charge ratio (1:1). The charge ratio (−/+) is the molar percentage of phosphate present in DNA and amine present in chitosan. (**II**) Chemical and structural analysis of the chitosan (CS) nanoparticles and the chitosan-GFP nanoparticles. (A,B) SEM images of CS nanoparticles at ×25,000 and ×10,000 magnification. (C,D) SEM images of CS-DNA nanoparticles at ×10,000 and ×2500 magnification at an accelerating voltage of 10 kV. (**III**) Size distribution spectra of the chitosan nanoparticles and the chitosan-DNA nanoparticles acquired using Dynamic Light Scattering. (A) Zeta-size of the chitosan nanoparticles showing two peaks: first peak at 313.3 nm and second peak at 96.34 nm. (B) Zeta size of the chitosan-DNA nanoparticles (289 ± 26.82 nm). (C) Zeta potential of the chitosan nanoparticles (28.8 mV) (D) Zeta potential of CS-DNA nanoparticles (10.6 mV).

**Figure 3 pharmaceuticals-17-00169-f003:**
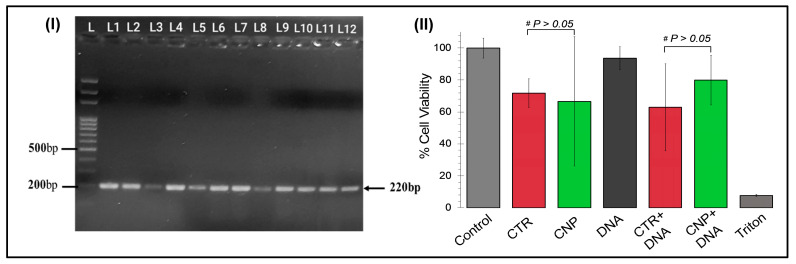
In vitro and in vivo assays to assess the safety and efficacy of the chemical transfection reagent (CTR) vs. the natural chitosan nanoparticles (CNP): (**I**) Representative gel electrogram confirming the GFP plasmid isolation via colony PCR. The L represents the ladder size (bp), whereas L1-L12 represents the PCR products of transformed colonies. (**II**) Representation of the percentage cell viability of U87 cell lines determined by in vitro cytotoxicity assay, post 48 h treatment with the chemical transfection reagent (CTR), chitosan nanoparticles (CNP), plasmid DNA alone, the CTR in combination with plasmid DNA (CTR + DNA), and the chitosan-GFP nanoparticles (CNP + DNA). Triton-X was used as a positive control for cell cytotoxicity. Data are expressed as mean ± SEM of three biological replicates. The chitosan and the chitosan-GFP formulations showed less cytotoxicity than the chemical transfection reagent CTR (*p* = 0.41; *p* = 0.24).

**Figure 4 pharmaceuticals-17-00169-f004:**
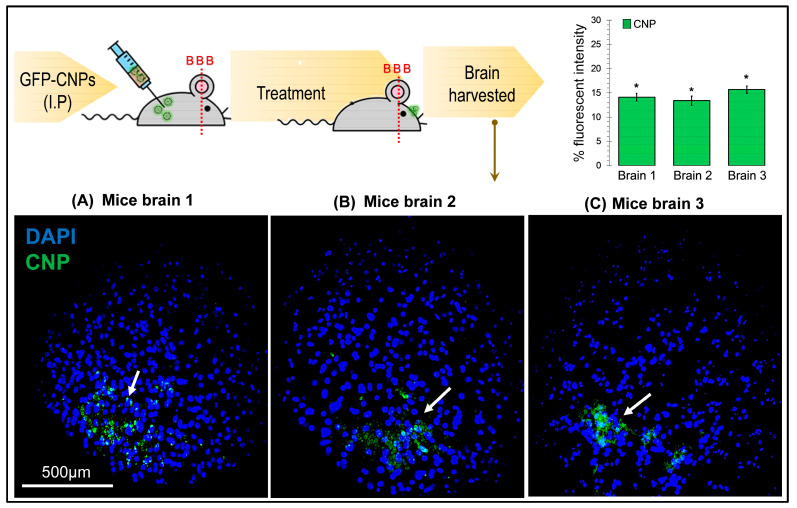
Representation of the immunofluorescence microphotographs (**A**–**C**) of 7-week-old male BALB/c mice that were injected with 200 μL of CS-GFP NPs intra-peritoneally to validate the entry of CS-GFP NPs through the BBB. The brains were carefully harvested following 24 h of treatment. No change was observed among the mean fluorescence intensity of the CNP in three different brains of the treated mice, *p* > 0.05 (* ANOVA: *p* = 0.197; N = 5; F crit (3.8) > F (1.8), indicating equal penetration of the CNP through the BBB (**B**).

**Figure 5 pharmaceuticals-17-00169-f005:**
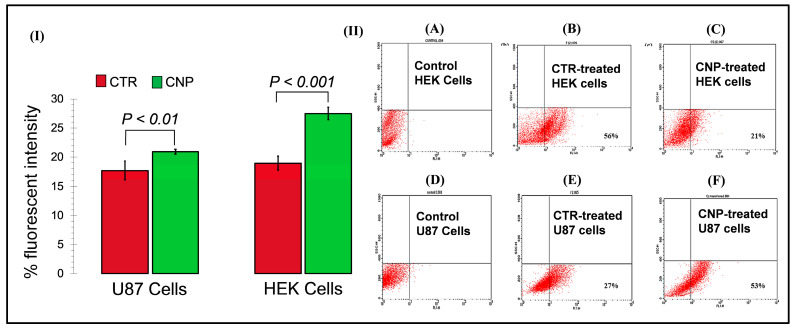
Evaluation of the transfection efficiency of the chitosan nanoparticles in HEK293- and glioma U87 cell lines: (**I**) Representation of the percent of transfected U87 and HEK293 cells following the CTR and CNPs treatment for 72 h, where the transfection rate was low with CTR (17.8% U87 cells; 18.9% HEK293 cells) as compared to the CNPs (20.6% U87 cells; 27.4% HEK293 cells). (**II**) The graphs represent the in vitro transfection efficiency of the CTR (B,E) and the CS-GFP NPs (C,F) on the HEK293 cells (A–C) and U-87 cells (D–F). The lower left quadrant represents the cell population with no GFP expression, whereas the lower right quadrant represents the GFP expression in the cells, measured via fluorescent-activated cell sorting following 24 h of incubation. Graphs (A,D) represent the cells-only controls. The total number of events recorded was 10,000.

**Figure 6 pharmaceuticals-17-00169-f006:**
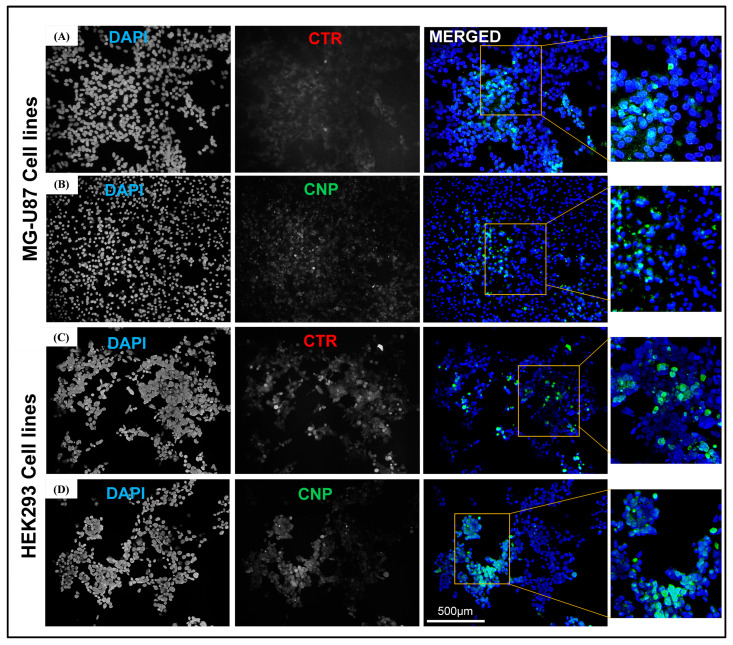
Representation of the microphotographs obtained via immunofluorescence microscopy following the treatment of the U87 and HEK293 cells with CTR (**A**,**C**) and CNPs (**B**,**D**). The blue color represents the nucleus stained with DAPI and the green represents the GFP tag.

## Data Availability

Data is contained within the article.

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
