# Peer review of "Chitosan-Based Polymeric Nanoparticles as an Efficient Gene Delivery System to Cross Blood Brain Barrier: In Vitro and In Vivo Evaluations"

_pharmaceuticals, 2024, doi:10.3390/ph17020169_

Round 1

Reviewer 1 Report

Comments and Suggestions for Authors

1. The discussion of size is contradictory. "The size of chitosan nanoparticles was found to be within the range of 89nm – 132nm. However, the size of the chitosan-GFP nanoparticles was ~260nm." "The average size distribution of the chitosan and chitosan-GFP nanoparticles was ~316 nm and 289 nm, respectively. The decrease in size after complex formation can be attributed to the size range for Chitosan-GFP nanoparticles is similar to the studies conducted by Zhao et al. and Fernandez et al."

2. How does this system cross the BBB? The authors should discuss. Breaking BBB or Transcytosis? (https://doi.org/10.1021/jacs.0c09029)

3. It is better to have some in vitro BBB model for studying brain targeting.

4. In the third part of "Discussion", the in vivo brain targeting should be discussed. Brain delivery is very challenging. This is novelty of this research.

5. The authors did not show any therapeutic results. The title should be modified.  It is better to remove "for the treatment of brain cancer". Otherwise, it is misleading.

Author Response

December 2, 2023

Editor-in-Chief, Pharmaceutics

I am sharing our revised version of our manuscript, pharmaceuticals-2730192, " Chitosan-based polymeric nanoparticles as an efficient gene delivery system to cross Blood brain barrier: In vitro and in vivo evaluations," for consideration for publication in Pharmaceutics.

We want to thank you, and the reviewers for your constructive feedback, which helped review the MS. Changes made are marked in blue.

We look forward to your decision on the suitability of this revised version for publication in pharmaceuticals.

Sincerely,

Dr. Aneela Javed

Dr. Muhammad Umar Aslam Khan

.........................................................................................................

List of Responses to Reviewers' Comments

Reviewer: 1

(Recommendation: Major Revision)

Reviewer Comment 1:

The discussion of size is contradictory. "The size of chitosan nanoparticles was found to be within the range of 89nm – 132nm. However, the size of the chitosan-GFP nanoparticles was ~260nm." "The average size distribution of the chitosan and chitosan-GFP nanoparticles was ~316 nm and 289 nm, respectively. The decrease in size after complex formation can be attributed to the size range for Chitosan-GFP nanoparticles is similar to the studies conducted by Zhao et al. and Fernandez et al."

Response to reviewer comment:

Thank you for the comments, the said justification has been provided as follows and manuscript has been updated.

These results are from two different techniques. SEM analysis shows the size and morphology of the nanoparticles but does not precisely represent the size of the whole batch of the nanoparticles prepared. However, Dynamic Light Scattering (DLS) analysis shows the size range of nanoparticles and the 1st peak in the figure 2 (III-A) small percentage of nanoparticles with average size of 96.34nm with 17.9% intensity can been seen and represent the small sample prepared for SEM analysis. The 2nd peak in DLS analysis shows an average size of 316.3nm with 82.1% intensity. So, the small percentage of nanoparticles with size range of 89nm – 132nm were observed in both SEM and DLS analysis but the percentage was less. In both cases the size is in nm and confirming the synthesis of Nano Particles.  In short SEM images provide rough estimation of particle size of the sample being observed compared to the sizes measured through the DLS technique. Therefore, having contradictory (but in range) sizes of NPs measured through both the techniques is understood as long as the size ranges are overlapping to a large extent, as in the present case.

Reviewer Comment 2:

How does this system cross the BBB? The authors should discuss. Breaking BBB or Transcytosis?

Response to reviewer comment:

Thank you for the comments, the said justification has been provided as follows and manuscript has been updated.

Pathophysiological barriers involving systemically administered nanomedicine can be divided functionally into (a) stimuli-responsive strategies (b) tissue microenvironment-reprogramming strategies (c) transcytosable nanomedicine; and (d) immuno-oncological nanomedicine [1]. Chitosan polymer offers advantages including muco-adhesion and functionalization ease. Chitosan-based nanocarriers (CsNCs) act by establishing ionic interactions with the endothelial cells thus accelerating the BBB crossing by adsorptive transcytosis. CsNCs can further be modified owed to their reactive amino and hydroxyl groups. Attachment/binding of ligands such as antibodies or lipids, CsNCs express a boosted passage across BBB [2].

References

1          Li, J.; Kataoka, K., Chemo-physical Strategies to Advance the in Vivo Functionality of Targeted Nanomedicine: The Next Generation. J. Am. Chem. Soc. 2021. 143. p. 538–559.

2          Caprifico, Anna E.; Foot, Peter J. S.; Polycarpou, E.; and Calabrese, G. Overcoming the Blood-Brain Barrier: Functionalised Chitosan Nanocarriers. Pharmaceutics. 2020. 12(11). p. 1013.

Reviewer Comment 3: 

It is better to have some in vitro BBB model for studying brain targeting.

Response to reviewer comment:

Thank you for the suggestion, In vitro is a basic approach and we have conducted the advanced assay, which is in vivo. Although the in vitro systems are available but as we did the in vivo experimentation which has successfully established the proof of concept. The in vitro assays are used where the in vivo experimentation is lacking or not possible etc. However, we will surely try to establish and utilize the in vitro assays for future studies currently we are collecting and arranging the resources and finances are available. The current manuscript will provide a basis for future grants in his regard.

Reviewer Comment 4: 

In the third part of "Discussion", the in vivo brain targeting should be discussed. Brain delivery is very challenging. This is novelty of this research.

Response to reviewer comment:

Neurological disorders affect more than one billion people worldwide and no effective therapies are available [1]. Several promising experimental drug candidates fail successive clinical trials due to insufficient delivery across BBB and subsequent expelling of therapeutic payload through efflux pumps (ABC transporters) leading towards severe side-effects such as diabetes, seizures, and toxicity [2]. Innovative technologies including nanocarriers (NC) with "enhanced permeability and retention" (EPR) are being explored. These therapies are meant to ensure targeted and efficient drug delivery across complex BBB for optimal therapeutic loading of gene/drug into the brain for treatment of Central Nervous System (CNS) disorders [3]. 

References

  1. Zheng, M.; et al. Nanotechnology-Based Strategies for siRNA Brain Delivery for Disease Therapy. Trends Biotechnol. 2018. 36(5). 562-575.
  2. Caprifico, Anna E.; Foot, Peter J. S.; Polycarpou, E. and Calabrese, G. Overcoming the Blood-Brain Barrier: Functionalised Chitosan Nanocarriers. Pharmaceutics. 2020. 12(11). 1013.
  3. Zhang, F.; Xu, C.L.; Liu, C.M. Drug delivery strategies to enhance the permeability of the blood–Brain barrier for treatment of glioma. Drug Des. Devel. Ther. 2015. 9. 2089–2100.

Reviewer Comment 5:

The authors did not show any therapeutic results. The title should be modified.  It is better to remove "for the treatment of brain cancer". Otherwise, it is misleading.

Response to reviewer comment:

Thank you very much for the suggestion. We have changed the title as below as suggested.

Chitosan-based polymeric nanoparticles as an efficient gene delivery system to cross Blood Brain Barrier: In vitro and in vivo evaluations

Reviewer 2 Report

Comments and Suggestions for Authors

1.) Abstract - The current problem statement in the abstract is generic. Abstract can be improved by specifying the problem statement by mentioning chitosan nanoparticles (CNPs) and Green Fluorescent Protein (GFP)-tagged plasmid.

2.) Abstract - A conclusion is required for the abstract. 

3.) Please standardize citation style. Eg. line 68 "and delivery of drugs to the brain [10] (Azad, et al., 2015).

4.) Introduction - Please describe the role of Green Fluorescent Protein (GFP) in treating cancer.

5.) Please provide a higher resolution FTIR spectra of pure gene, NP and NP with gene. 

6.) The SEM figure is not clear as well. It looks more like a TEM results. Please provide higher resolution SEM results. 

7.) Suggest to split the results into individual figures. Now 3-4 types results are combined in a single figure and this affects the resolution. 

8.) Please provide the complete information of the materials and equipment used eg the supplier name and country. For the equipment, the model number is required.

9.) Is the in-vivo study approved by any animal ethics board? Please provide the information

Comments on the Quality of English Language

Moderate editing of English language required

Author Response

December 2, 2023

Editor-in-Chief, Pharmaceutics

I am sharing our revised version of our manuscript, pharmaceuticals-2730192, " Chitosan-based polymeric nanoparticles as an efficient gene delivery system to cross Blood brain barrier: In vitro and in vivo evaluations," for consideration for publication in Pharmaceutics.

We want to thank you, and the reviewers for your constructive feedback, which helped review the MS. Changes made are marked in blue.

We look forward to your decision on the suitability of this revised version for publication in pharmaceuticals.

Sincerely,

Dr. Aneela Javed

Dr. Muhammad Umar Aslam Khan

.........................................................................................................

Reviewer: 2

(Recommendation: Major Revision)

Reviewer Comment 1:

Abstract - The current problem statement in the abstract is generic. Abstract can be improved by specifying the problem statement by mentioning chitosan nanoparticles (CNPs) and Green Fluorescent Protein (GFP)-tagged plasmid.

Response to reviewer comment:

The whole abstract has been revised with said suggestion as follows and manuscript has been updated.

Gene therapy has made significant progress, but effective treatments for brain tumors remain challenging due to their complex nature. Current treatment options have limitations, specifically precisely the inability to cross the blood-brain barrier (BBB) and precisely target cancer cells. Therefore, alternative treatment options that are safer, more effective, and capable of specifically targeting cancer cells are urgently required as an alternative. Current study aimed. We have developed highly biocompatible natural biopolymeric chitosan nanoparticles (CNPs) as a potential gene delivery vehicle that can cross the BBB. The current study aimed to develop highly biocompatible natural biopolymeric chitosan nanoparticles (CNPs) as a potential gene delivery vehicle that can cross the BBB and serve as a gene or drug delivery vehicle for brain diseases therapeutics., which are highly biocompatible natural biopolymers. The efficiency of the CNPs was evaluated via in-vitro transfection of Green Fluorescent Protein (GFP)-tagged plasmid in HEKHEK293-293 and brain cancer MG-U87 cell lines, as well as in in-vivo mouse models. The CNPs were prepared via a complex coacervation method, resulting in nanoparticles of approximately 260nm in size. In-vitro cytotoxicity analysis revealed that CNPs had better cell viability (85%) in U87 cells compared to the chemical transfection reagent (CTR) (72%). Moreover, transfection efficiency of CNPs was also found to be transfection efficiency of CNPs was also higher, as indicated by fluorescent emission microscopy (20.56% vs 17.79%) and fluorescent activated cell sorting (53% vs 27%). In-vivo assays using Balb/c mice revealed that CNPs could efficiently cross the BBB, suggesting their potential as an efficient gene delivery vehicle for targeted therapies against brain cancers as well as other brain diseases where efficient targeting of therapeutic load to the brain cells has proven to be a real challenge.

Reviewer Comment 2:

Abstract - A conclusion is required for the abstract.

Response to reviewer comment:

The whole abstract has been revised with said suggestion as follows and manuscript has been updated.

Gene therapy has made significant progress, but effective treatments for brain tumors remain challenging due to their complex nature. Current treatment options have limitations, specifically precisely the inability to cross the blood-brain barrier (BBB) and precisely target cancer cells. Therefore, alternative treatment options that are safer, more effective, and capable of specifically targeting cancer cells are urgently required as an alternative. Current study aimed. We have developed highly biocompatible natural biopolymeric chitosan nanoparticles (CNPs) as a potential gene delivery vehicle that can cross the BBB. The current study aimed to develop highly biocompatible natural biopolymeric chitosan nanoparticles (CNPs) as a potential gene delivery vehicle that can cross the BBB and serve as a gene or drug delivery vehicle for brain diseases therapeutics., which are highly biocompatible natural biopolymers. The efficiency of the CNPs was evaluated via in-vitro transfection of Green Fluorescent Protein (GFP)-tagged plasmid in HEKHEK293-293 and brain cancer MG-U87 cell lines, as well as in in-vivo mouse models. The CNPs were prepared via a complex coacervation method, resulting in nanoparticles of approximately 260nm in size. In-vitro cytotoxicity analysis revealed that CNPs had better cell viability (85%) in U87 cells compared to the chemical transfection reagent (CTR) (72%). Moreover, transfection efficiency of CNPs was also found to be transfection efficiency of CNPs was also higher, as indicated by fluorescent emission microscopy (20.56% vs 17.79%) and fluorescent activated cell sorting (53% vs 27%). In-vivo assays using Balb/c mice revealed that CNPs could efficiently cross the BBB, suggesting their potential as an efficient gene delivery vehicle for targeted therapies against brain cancers as well as other brain diseases where efficient targeting of therapeutic load to the brain cells has proven to be a real challenge.

Reviewer Comment 3: 

Please standardize citation style. Eg. line 68 "and delivery of drugs to the brain [10] (Azad, et al., 2015).

Response to reviewer comment:

The citation style was aligned with the required reference style and manuscript has been updated.

Reviewer Comment 4: 

Introduction - Please describe the role of Green Fluorescent Protein (GFP) in treating cancer.

Response to reviewer comment:

Thanks for the suggestion. The required information has been incorporated as followed in manuscript line as follows and manuscript has been updated.

"GFP was used as a control as it can be easily visualized and tracked for in vitro transfections crossing cell membrane and also in vivo crossing the BBB through various techniques utilized in studies including FACS and fluorescent microscopy.?

Reviewer Comment 5:

Please provide a higher resolution FTIR spectra of pure gene, NP and NP with gene.

Response to reviewer comment:

Thank you very much for the suggestion and we have provided the high resolution FTIR spectra of the product as follows and manuscript has been updated.

Reviewer Comment 6:

The SEM figure is not clear as well. It looks more like a TEM results. Please provide higher resolution SEM results.

Response to reviewer comment:

Thank you very much for the suggestion and we have provided the high-resolution SEM spectra of the product as follows and manuscript has been updated.

Reviewer Comment 7:

Suggest to split the results into individual figures. Now 3-4 types results are combined in a single figure and this affects the resolution.

Response to reviewer comment:

Thank you very much for the suggestion and we have provided the high-resolution images and split images as follows and manuscript has been updated.

Figure 2. Characterization of Chitosan-GFP Nanoparticles: (I) FTIR spectrum of Chitosan-GFP Nanoparticles at charge ratio (1:1). Charge ratio (-/+) is the molar percentage of phosphate present in DNA and amine present in Chitosan. (II) Chemical and structural analysis of the chitosan (CS) nanoparticles and chitosan-GFP nanoparticles. (A, B) SEM images of CS nanoparticles at x25,000 and x10,000 magnification. (C, D) SEM images of CS-DNA nanoparticles at x10,000 and x2,500 magnification at an accelerating voltage of 10kV. (III) Size distribution spectra of chitosan nanoparticles and chitosan-DNA nanoparticles acquired through Dynamic Light Scattering. (A) Zeta-size of chitosan nanoparticles showing two peaks, 1st peak at 313.3 nm and 2nd peak at 96.34nm. (B) Zeta size of chitosan-DNA nanoparticles (289 ± 26.82nm). (C) Zeta potential of chitosan nanoparticles (28.8mV) (D) Zeta potential of CS-DNA nanoparticles (10.6mV).

Figure 3: In-vitro and In-vivo assays to assess the safety and efficacy of chemical transfection reagent (CTR) vs natural chitosan nanoparticles (CNP): (3-I) Representative gel electrogram confirming the GFP plasmid isolation via colony PCR. The L represents the ladder size (bp), whereas L1-L12 represents the PCR products of transformed colonies. (3-II) Represent percentage cell viability of U87 cell lines determined by in-vitro cytotoxicity assay, following 48 hours of treatment with chemical transfection reagent (CTR), chitosan nanoparticles (CNP), plasmid DNA alone, CTR in combination with plasmid DNA (CTR + DNA), and chitosan-GFP nanoparticles (CNP + DNA). Triton-X was used as a positive control for cell cytotoxicity. Data is expressed as mean ± SEM of three biological replicates. Chitosan and the chitosan-GFP formulation showed less cytotoxicity than the chemical transfection reagent CTR (p = 0.41; p = 0.24).

Figure 4 Represents the immunofluorescence microphotographs (A, B & C) of 7 weeks male BALB/c mice that were injected with the 200ul of CS-GFP NPs intra-peritoneally to validate the entry of CS-GFP NPs through the BBB and brains were carefully harvested following 24 hours of treatment. No change was observed among the mean fluorescence intensity of CNP in three different brains of treated mice, P >0.05 (*ANOVA: p = 0.197; N = 5; F crit (3.8) > F (1.8), indicating equal penetration of CNP through the BBB (4-B).

Figure 5: Evaluation of transfection efficiency of Chitosan Nanoparticles in HEK293- and glioma U87 cell lines: (I) Represent the percent of transfected U87 and HEK293 cells following CTR and CNP's treatment for 72 hours, where the transfection rate was low with CTR (17.8% U87 cells; 18.9% HEK293 cells) compared to the CNPs (20.6% U87 cells; 27.4% HEK293 cells). II) The graphs represent the in-vitro transfection efficiency of CTR (B, E) and CS-GFP NPs (C, F) on HEK293 cells (A, B & C) and U-87 cells (D, E & F). The lower left quadrant represents the cell population with no GFP expression, whereas the lower right quadrant represents the GFP expression in the cells, measured via fluorescent-activated cell Sorting following 24 hours of incubation. Graphs A & D represent the cells only controls. The total number of events recorded was 10,000.

 Figure 6: Represent the microphotographs obtained via immunofluorescence microscopy following treatment of U87 and HEK293 cells with CTR (A, C) and CNPs (B, D), where the blue color represents the nucleus stained with DAPI and green represents the GFP tag.

Reviewer Comment 8:

Please provide complete information of the materials and equipment used eg the supplier name and country. For the equipment, the model number is required.

Response to reviewer comment:

Thank you very much for the suggestion and have added names of companies and suppliers etc where applicable in material and method section and manuscript has been updated.

Reviewer Comment 9:

Is the in-vivo study approved by any animal ethics board? Please provide the information

Response to reviewer comment:

The in vivo study was approved by ethical board as follows and manuscript has been updated.

As mentioned in ethical statement "This study was approved by the Institutional Review Board of Atta-ur-Rahman School of Applied Biosciences (ASAB), National University of Sciences and Technology (NUST). All procedures on animals were performed in compliance with the rulings of the Institute of Laboratory Animal Research, Division on Earth and Life Sciences, National Institutes of Health, USA (Guide for the Care and Use of Laboratory Animals: Eighth Edition, 2011). "The IRB number for the approval is 04-2021-01/02

Reviewer 3 Report

Comments and Suggestions for Authors

The manuscript “ Chitosan-based polymeric nanoparticles as an efficient gene delivery system for the treatment of brain cancer: An in vitro and  in vivo evaluations’ submitted  by Ishaq N. Khan, et al. the paper The study outlined the development and evaluation of chitosan nanoparticles (CNPs) as a potential gene delivery vehicle for the treatment of brain tumors. The main challenges addressed in the study include the complex nature of brain tumors, limitations of current treatment options in crossing the blood-brain barrier (BBB), and the need for safer and more effective alternatives. The study suggests that chitosan nanoparticles (CNPs) hold promise as a gene delivery vehicle for brain tumor treatment. Their biocompatibility, size, and demonstrated ability to cross the blood-brain barrier make them a potential candidate for targeted therapies against brain cancers. The findings emphasize the need for alternative treatments that can address the challenges associated with current options for brain tumor therapy.

The very first thing I want to say is that this is a very good and necessary manuscript. And this article is well written, logical, well-illustrated and contains many interesting facts.  The topic of research is relevant and I recommended acceptance after addressing some minor comments.

1.     Line 49, define the abbreviation GBM as it was first mention and LGG not defined in the the paper.

2.     Line 68, (Azad et al. 2015) no need to mention it delete it because ref. number mentioned.

3.     The references are not uniform, do it in journal style format

Author Response

December 2, 2023

Editor-in-Chief, Pharmaceutics

I am sharing our revised version of our manuscript, pharmaceuticals-2730192, " Chitosan-based polymeric nanoparticles as an efficient gene delivery system to cross Blood brain barrier: In vitro and in vivo evaluations," for consideration for publication in Pharmaceutics.

We want to thank you, and the reviewers for your constructive feedback, which helped review the MS. Changes made are marked in blue.

We look forward to your decision on the suitability of this revised version for publication in pharmaceuticals.

Sincerely,

Dr. Aneela Javed

Dr. Muhammad Umar Aslam Khan

.........................................................................................................

List of Responses to Reviewers' Comments

Reviewer: 3

(Recommendation: Major Revision)

Reviewer Comment 1:

Line 49, define the abbreviation GBM as it was first mention and LGG not defined in the paper.

Response to reviewer comment:

The complete names have been added along with abbreviations. GBM (glioblastoma multiforme) and LGGs (Lower grade Gliomas)

Reviewer Comment 2:

Line 68, (Azad et al. 2015) no need to mention it delete it because ref. number mentioned.

Response to reviewer comment:

The citation style was aligned with the required reference style.

Reviewer Comment 3: 

The references are not uniform, do it in journal style format

Response to reviewer comment:

The format for references has been corrected.

Round 2

Reviewer 1 Report

Comments and Suggestions for Authors

"Pathophysiological barriers involving systemically administered nanomedicine can be divided functionally into (a) stimuli-responsive strategies (b) tissue microenvironment-reprogramming strategies (c) transcytosable nanomedicine; and (d) immuno-oncological nanomedicine." Please check this sentence. Pathophysiological barriers....can be divided...?  It is wrong.

Author Response

Reviewer: 1

(Recommendation: Major Revision)

Reviewer Comment 1:

"Pathophysiological barriers involving systemically administered nanomedicine can be divided functionally into (a) stimuli-responsive strategies (b) tissue microenvironment-reprogramming strategies (c) transcytosable nanomedicine; and (d) immuno-oncological nanomedicine." Please check this sentence. Pathophysiological barriers....can be divided...?  It is wrong.

Response to reviewer comment:

Thank you for highlighting the correction, the said suggestion has been addressed as follows and manuscript has been updated.

systemically administered nanomedicine strategies can be divided functionally into (a) stimuli-responsive strategies, (b) tissue microenvironment-reprogramming strategies, (c) transposable nanomedicine, and (d) immuno-oncological nanomedicine.

Reviewer 2 Report

Comments and Suggestions for Authors

The authors have addressed all my comments.

Author Response

Thank you for the feedback that helped to improve the our manuscript. The manuscript have been revised by field expert to resolve English language and other technical issues.